# Enhancing LLM Reliability via Explicit Knowledge Boundary Modeling

**Hang Zheng**[1,2*]  **Hongshen Xu**[1,2*]  **Yuncong Liu**[1,2]  **Shuai Fan**[2,3]
**Pascale Fung**[4]  **Lu Chen**[1,2,5†]  **Kai Yu**[1,2,5†]
[1]X-LANCE Lab, School of Computer Science
MoE Key Lab of Artificial Intelligence, SJTU AI Institute
Shanghai Jiao Tong University, Shanghai, China
[2]Jiangsu Key Lab of Language Computing, Suzhou, China
[3]AISpeech Co., Ltd., Suzhou, China
[4]Center for Artificial Intelligence Research (CAiRE)
Hong Kong University of Science and Technology, Hong Kong, China
[5]Suzhou Laboratory, Suzhou, China
{azure123, xuhongshen, chenlusz, kai.yu}@sjtu.edu.cn

## Abstract

Large language models (LLMs) are prone to hallucination stemming from misaligned self-awareness, particularly when processing queries exceeding their knowledge boundaries. While existing mitigation strategies employ uncertainty estimation or query rejection mechanisms, they suffer from computational efficiency and sacrificed helpfulness. To address these issues, we propose the Explicit Knowledge Boundary Modeling (*EKBM*) framework, integrating fast and slow reasoning systems to harmonize reliability and usability. The framework first employs a fast-thinking model to generate confidence-labeled responses, enabling immediate utilization of high-confidence outputs, whereas uncertain predictions trigger a slow refinement model for accuracy improvement. To align model behavior with our proposed object, we propose a hybrid training pipeline, enhancing self-awareness without degrading task performance. Evaluations on dialogue state tracking tasks demonstrate that *EKBM* achieves superior model reliability over uncertainty-based baselines. Further analysis reveals that refinement substantially boosts accuracy while maintaining low computational overhead. The framework establishes a scalable paradigm for deploying reliable LLMs in error-sensitive applications, effectively balancing accuracy and practical utility.

## 1 Introduction

Recently, large language models (LLMs) have demonstrated impressive text generation capabilities (Abdullah et al., 2022). However, LLMs are susceptible to hallucinations, where generated content misaligns with context or factual information (Zhang et al., 2023). Hallucinations can be particularly detrimental in applications with low tolerance for error, eroding user trust in LLM reliability.

Extensive works have been devoted to mitigating hallucinations in LLMs (Tonmoy et al., 2024). Hallucinations often arise from misalignments between an LLM's outputs and its intrinsic knowledge boundaries, particularly when when addressing queries beyond its expertise (Li et al., 2024b). Improving the model's self-awareness—its capacity to accurately assess its own knowledge boundaries and outputs correctness, can effectively mitigate this misalignment. Uncertainty-based methods estimate model confidence and indirectly

---

*Hang Zheng and Hongshen Xu contribute equally to this work.
†Lu Chen and Kai Yu are the corresponding authors.

reflect the model's self-awareness (Huang et al., 2024), facing challenges of inherence to the model's endogenous capabilities, frequently sufferring from high computational costs or instability across thresholds. Alternative strategies emphasize aligning knowledge boundaries with actions by rejection of out-of-scope queries (Xu et al., 2024b), avoiding mistakes but sacrificing helpfulness—a trade-off that undermines model utility particularly in complex, multi-step tasks where partial correctness remains valuable.

To address these limitations, we propose a novel alignment objective that enhances model awareness of its knowledge boundaries, enabling models to respond more effectively by explicitly distinguishing between high-confidence ("sure") and low-confidence ("unsure") outputs. A reliable system should deliver near-perfect accuracy for "sure" predictions and provide helpful information in the "unsure" category, serving as a form of soft rejection, appropriately managing user expectations regarding accuracy.

Building on this concept, we propose the Explicit Knowledge Boundary Modeling (*EKBM*) framework. *EKBM* integrates both Fast and Slow systems: a fast-thinking model generates responses annotated with confidence labels, allowing for immediate use of high-confidence outputs, while a slow-thinking refinement model engages in deliberate reasoning to improve low-confidence predictions. Crucially, *EKBM* does not merely filter uncertain outputs but leverages them as opportunities for improvement, ensuring both precision and recall. The collaboration between these two systems strikes a balance between reliability and efficiency, with the fast-thinking model's self-awareness being crucial for the system's effective functioning.

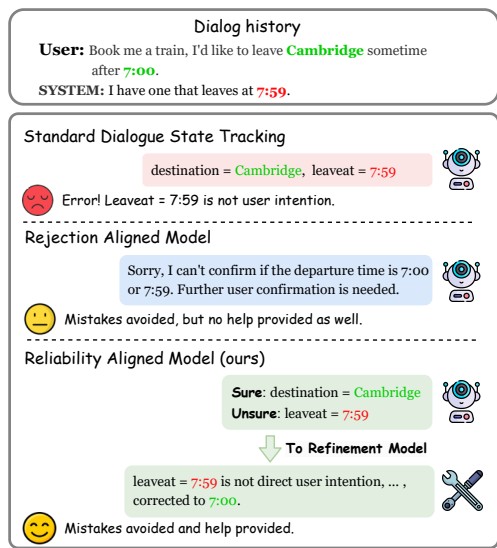

Figure 1: A case study on Dialog State Tracking: comparison of different alignment objectives.

To operationalize this paradigm, we design a training pipeline that aligns the model's capability boundaries and improves self-awareness. Evaluations on dialogue state tracking (DST) tasks demonstrate that our method effectively and reliably distinguishes the confidence levels of model output compared to baseline methods. Additionally, we develop an automatic refinement model for "unsure" outputs. Experiments show that the *EKBM* system achieves superior overall performance compared to traditional approaches that directly tune models for the task.

Our contributions are summarized as follows:

- We present the *EKBM* framework, integrating fast-thinking assessment with slow-thinking refinement, along with metrics for evaluating self-awareness and reliability.
- We develop a comprehensive training pipeline that enhances LLM self-awareness, yielding more reliable models.
- We perform extensive experiments to demonstrate the effectiveness and scalability of the *EKBM* framework, providing valuable insights for future research.

## 2 Problem Formulation

### 2.1 Alignment for Reliability

To enhance the reliability of LLMs, a prevailing method involves aligning their performance with high-quality training data. This alignment aims to achieve the following objective:

$$s(x, y_c) > s(x, y_w) \tag{1}$$

where a scoring function $s(x, y)$ ensures that the score of a correct response $y_c$ to an input prompt $x$ is higher than that of an incorrect response $y_w$.

Recent studies have introduced rejection as a mechanism to help the model differentiate between answerable (in-boundary) and unanswerable (out-of-boundary) queries, where model is encouraged to directly reject the unanswerable ones. The corresponding alignment objective can be articulated as:

$$s(x, y_c) > s(x, y_r) > s(x, y_w) \tag{2}$$

where $y_r$ denotes a truthful rejection, and $y_w$ an incorrect response. This suggests that refusal is preferable to providing an incorrect answer.

While this rejection strategy ensures reliability, it may unduly limit the model's helpfulness. Therefore, we propose a more nuanced objective: the model should provide an answer whenever possible, categorizing its output as either high-confidence ("sure") or low-confidence ("unsure"). The model must ensure high accuracy for "sure" responses while tolerating some errors in the "unsure" category, which can serve as references for users or inputs for further refinement. Our proposed alignment objective is formalized as:

$$s(x, y_c, c_s) > s(x, y_c, c_u) > s(x, y_w, c_u) > s(x, y_w, c_s) \tag{3}$$

Here, $y_c$ and $y_w$ denote correct and incorrect responses, while $c_s$ and $c_u$ represent "sure" and "unsure" confidence levels. This objective guarantees that correct "sure" predictions are prioritized over correct "unsure" predictions, which, while less confident, still provide utility. Incorrect "unsure" predictions are discouraged but considered more acceptable than incorrect "sure" predictions, as the latter significantly undermine overall reliability.

## 2.2 Reliability Evaluation

In this work, we focus on complex multi-slot problems and have modified several metrics for better evaluation of model reliability.

### 2.2.1 Weighted-F1

The F1-score, calculated using True Positives (TP), False Positives (FP), and False Negatives (FN), is typically the evaluation metric for multi-slot problems:

$$\text{Precision} = \frac{\text{TP}}{\text{TP} + \text{FP}}, \quad \text{Recall} = \frac{\text{TP}}{\text{TP} + \text{FN}} \tag{4}$$

$$\text{F1} = 2 \cdot \frac{\text{Precision} \cdot \text{Recall}}{\text{Precision} + \text{Recall}} \tag{5}$$

We modify the F1 score to align with the objectives of our *EKBM* framework. Since "unsure" predictions represent an state of uncertainty, we assign a weighted contribution (denoted by $\alpha_1$ and $\alpha_2$, both ranging from 0 to 1) to reflect their partial utility as candidates for refinement. The modified precision and recall are defined as:

$$\text{Precision}(\alpha_1) = \frac{\text{STP} + \alpha_1 \cdot \text{UTP}}{\text{STP} + \text{SFP} + \alpha_1 \cdot (\text{UTP} + \text{UFP})}, \quad \text{Recall}(\alpha_2) = \frac{\text{STP} + \alpha_2 \cdot \text{UTP}}{\text{STP} + \text{UTP} + \text{FN}} \tag{6}$$

Here, STP and SFP denote True Positives and False Positives for "sure" predictions, while UTP and UFP are the corresponding values for the "unsure" category. For recall, we consider a prediction to be a UTP if it is partially correct (i.e., refinable). Unlike precision, we do not apply the weighting parameter $\alpha_2$ to the denominator of the recall calculation, as recall measures the proportion of gold labels retrieved, which is independent of the introduction of $\alpha$. Notably, incorporating $\alpha$ does not affect the range of precision and recall, both of which remain within [0,1]. The modified **Weighted-F1** score is defined as:

$$\text{Weighted-F1}(\alpha_1, \alpha_2) = 2 \cdot \frac{\text{Precision}(\alpha_1) \cdot \text{Recall}(\alpha_2)}{\text{Precision}(\alpha_1) + \text{Recall}(\alpha_2)} \tag{7}$$

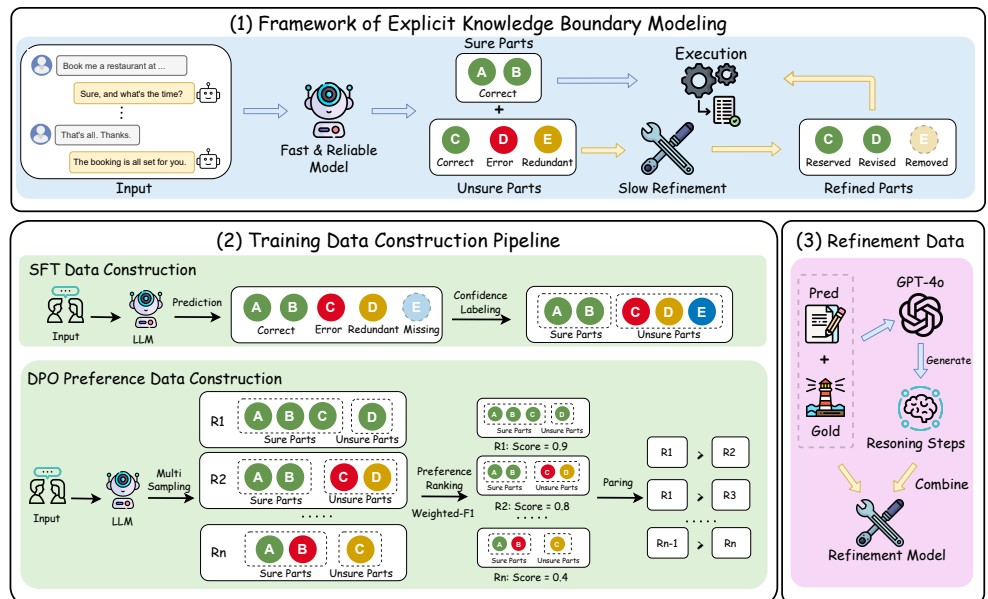

Figure 2: **The *EKBM* framework and the data construction methods.**

### 2.2.2 *Metric-Objective Consistency: Weighted-F1 as a Parametric Alignment Measure*

The Weighted-F1 metric aligns with our four-tiered objective (Equation (3)) by using a parametric formulation that encodes hierarchical priorities, which ensures: (1) **Strict enforcement for sure predictions**: Unweighted terms prioritize high-confidence correctness (STP) and apply maximal penalties for errors (SFP). (2) **Controlled utility for unsure predictions**: The $\alpha$-scaled terms modulate low-confidence outputs, with attenuated UFP penalties tolerating unsure errors, and partial UTP rewards incentivizing refinable correct predictions.

This dual-parameter mechanism enables scenario-specific adaptation: Weighted-Precision regulates error tolerance via $\alpha_1$ (higher $\alpha_1$ tightens unsure-error constraints) and Weighted-Recall controls reference-value incentives via $\alpha_2$ (higher $\alpha_2$ promotes helpfulness through unsure-correct outputs). The composite metric thus operationalizes the reliability-helpfulness trade-off, permitting calibration from conservative ($\alpha_1 \rightarrow 1$, $\alpha_2 \rightarrow 0$) to exploratory ($\alpha_1 \rightarrow 0$, $\alpha_2 \rightarrow 1$) strategies.

We discuss a few corner cases: (1) $\alpha_1 = 0$, $\alpha_2 = 0$: Weighted-F1 disregards all "unsure" predictions. (2) $\alpha_1 = 0$, $\alpha_2 = 1$: Errors in the "unsure" category are ignored, rewarding effective recall, representing the theoretical upper limit after perfect refinement. (3) $\alpha_1 = 1$, $\alpha_2 = 0$: Lacks significant interpretation. (4) $\alpha_1 = 1$, $\alpha_2 = 1$: Eliminates confidence labeling, reverting to traditional methods.

### 2.2.3 *Reliability Metrics*

We use two specific Weighted-F1 scores: Quality-F1 as the primary metric for assessing model reliability, and Optimal-F1 as a supplementary reference.

**Quality-F1**: Defined as Weighted-F1$(0.5, 0.5)$, this metric assigns half the weight to "unsure" predictions compared to "sure" ones. This encourages the model to provide high-confidence "sure" outputs while mitigating penalties for "unsure" ones. Quality-F1 directly measures the model's self-awareness and reliability.

**Optimal-F1** Defined as Weighted-F1$(0, 1)$, it represents the theoretical upper limit achievable after perfect refinement of "unsure" predictions, reflecting the overall potential of the system.

## 3 Method

### 3.1 Explicit Knowledge Boundary Modeling

We formalize our proposed framework, Explicit Knowledge Boundary Modeling (*EKBM*), which aims to improve model reliability and self-awareness by explicitly modeling its

knowledge boundaries and improve overall performance by incorporating a refinement model to improve the "unsure" predictions. As shown in Figure 2 (1), the framework divides the model's decision-making process into two distinct stages, balancing immediate usability with comprehensive coverage, making the system both reliable and helpful. We provide a complete example of the two-stage execution flow and model outputs in Appendix B.

**Fast Prediction with Confidence Labeling**   In the first stage, the model performs the primary task and simultaneously assigns a confidence label ("sure" or "unsure") to each prediction. "Sure" predictions are directly accepted as final outputs, ensuring high precision.

**Slow Refinement for Unsure Prediction**   In the second stage, "unsure" predictions undergo further refinement through strategies like user confirmation, automated multi-step reasoning, or other post-processing techniques, incuring extra cost but increases overall accuracy and reliability of the system.

### 3.2   Reliability Alignment

To align the model's behavior with our alignment objective, we explore various training strategies to enhance its intrinsic self-awareness (Xu et al., 2025), including Supervised Fine-Tuning (SFT) and Direct Preference Optimization (DPO). The data construction process is shown in Figure 2(2), with two detailed examples provided in Appendix §C.

**SFT Data Construction**   We assume that a model's task-specific knowledge boundary can be approximated by repeated sampling on training data. Accordingly, our SFT data construction is straightforward: for each training sample, we perform multiple samplings. Predictions consistently deemed accurate are labeled "sure," while those with errors or omissions are marked "unsure." Crucially, we intentionally **refrain from correcting erroneous outputs** to avoid introducing supervisory signals that might inadvertently improve the model's task performance. That is, when the model produces an incorrect prediction, we simply assign a confidence label of "unsure" without rectifying the error. The erroneous output remains in the final training data, reinforcing the model's ability to recognize and categorize uncertain information rather than improving its accuracy directly.

The number of sampling rounds $i$ is a hyperparameter. A larger $i$ results in a more conservative model, as the proportion of "unsure" labels in the training corpus increases. In our study, we set $i = 1$ by default to accurately reflect the model's realistic capability boundary.

**DPO Preference Data Construction**   DPO enables the model to learn directly from preference pairs, making the design of an effective preference strategy crucial. Given our alignment objective, it is intuitive to use Weighted-F1 as the preference score function. The $\alpha$ values in Weighted-F1 significantly influence the model's behavior and performance. Through experimentation (see §4.5), we determined that setting $\alpha_1$ to 0.25 and $\alpha_2$ to 0.75 best balances the model's immediate helpfulness and its potential performance after refinement. Therefore, this setting will be used by default unless specified. To enhance the stability of DPO training, we ensure that all data is generated through multi-sampling from the model itself, aligning with the model's output distribution.

### 3.3   Refinement Model

We train automated refinement models for each dataset to further optimize unsure predictions by multi-step reasoning, leveraging supervised fine-tuning (SFT) based on the LLAMA3 8B model. Our refinement operates at a fine-grained level, addressing each unsure prediction individually. We employ a reasoning paradigm similar to the Chain of Thought (Wei et al., 2022) methodology and use GPT-4o (Hurst et al., 2024) to generate detailed reasoning process for each training prediction. This approach is illustrated in Figure 2 (3), and detailed information can be found in Appendix §D.

## 4   Experiments

We conduct experiments focusing on three core research questions:

**RQ1**   Does our training pipeline outperform current methods in enhancing self-awareness?

**RQ2**   Does *EKBM* framework improve the overall performance of LLMs on complex tasks?

**RQ3**   Does our approach scale robustly across foundation models with varying capabilities?

| Method Type | Method | MultiWOZ-2.4 | | | | BiTOD | | | | SGD | | | |
|---|---|---|---|---|---|---|---|---|---|---|---|---|---|
| | | Prec$_{sure}$ ↑ | Rec$_{total}$ ↑ | Opti. F1 ↑ | Qual. F1 ↑ | Prec$_{sure}$ ↑ | Rec$_{total}$ ↑ | Opti. F1 ↑ | Qual. F1 ↑ | Prec$_{sure}$ ↑ | Rec$_{total}$ ↑ | Opti. F1 ↑ | Qual. F1 ↑ |
| *LLAMA3-8B* Dubey et al. (2024) | | | | | | | | | | | | | |
| Prompt | Direct | 72.48 | 77.98 | 73.38 | 73.38 | 73.37 | 64.83 | 65.39 | 65.39 | 26.46 | 28.71 | 25.12 | 25.12 |
| | Verbose | 85.95 | 85.85 | 80.88 | 53.34 | 91.53 | 76.52 | 78.09 | 53.56 | 69.59 | 27.93 | 29.25 | 14.98 |
| | SR | 86.43 | 81.34 | 81.34 | 61.43 | 92.39 | 76.32 | 78.84 | 48.60 | 65.32 | 33.61 | 33.37 | 21.30 |
| Uncertainty | Prob | 80.05 | 80.66 | 77.74 | 69.03 | 83.50 | 72.44 | 73.40 | 60.30 | 36.93 | 32.28 | 27.74 | 22.86 |
| | SC | 80.11 | 89.51 | 84.15 | 64.48 | 83.16 | 75.48 | 77.33 | 62.93 | 84.66 | 41.06 | 42.76 | 16.55 |
| | P(True) | 72.82 | 79.00 | 73.50 | 70.81 | 72.44 | 66.26 | 65.97 | 63.06 | 24.78 | 29.93 | 26.26 | 26.24 |
| Reliability (Ours) | SFT | 94.04 | 91.75 | 91.65 | 82.44 | 95.66 | 88.25 | 88.56 | 83.08 | 86.74 | 85.51 | 82.66 | 61.64 |
| | + DPO Joint | **94.52** | 92.16 | **92.83** | 83.50 | 95.84 | 88.85 | 88.89 | 83.11 | **89.07** | **86.31** | **85.89** | 61.42 |
| | + DPO Post | 93.59 | **93.10** | 91.75 | **83.55** | 95.17 | 88.00 | 88.35 | 85.78 | 82.26 | 83.47 | 79.70 | **64.14** |
| *Qwen2.5-7B* Yang et al. (2024) | | | | | | | | | | | | | |
| Prompt | Direct | 69.42 | 71.37 | 67.22 | 67.22 | 71.19 | 65.31 | 65.98 | 65.98 | 33.43 | 36.11 | 33.40 | 33.40 |
| | Verbose | 83.02 | 71.14 | 69.05 | 58.88 | 78.62 | 77.22 | 75.23 | 64.78 | 65.78 | 31.80 | 32.13 | 21.74 |
| | SR | 73.54 | 72.08 | 69.69 | 67.87 | 73.73 | 66.48 | 67.90 | 66.34 | 35.87 | 36.82 | 35.13 | 34.05 |
| Uncertainty | Prob | 77.12 | 74.47 | 72.02 | 65.19 | 84.57 | 74.43 | 75.49 | 61.11 | 46.66 | 38.91 | 34.83 | 29.36 |
| | SC | 83.33 | 90.20 | 84.67 | 57.00 | 87.65 | 74.83 | 77.42 | 51.43 | 71.47 | 61.08 | 53.38 | 25.22 |
| | P(True) | 70.03 | 71.44 | 67.49 | 67.24 | 72.11 | 65.66 | 66.13 | 65.65 | 33.60 | 36.40 | 33.67 | 33.56 |
| Reliability (Ours) | SFT | 94.16 | 91.93 | 91.98 | 82.03 | 96.08 | 89.13 | 89.33 | 83.01 | 90.69 | 83.86 | 83.09 | 64.63 |
| | + DPO Joint | **95.29** | **92.88** | **93.03** | 80.86 | 95.99 | **89.86** | 89.51 | 83.01 | **91.23** | **85.78** | **83.95** | 64.48 |
| | + DPO Post | 94.02 | 92.23 | 92.12 | **82.89** | **96.15** | 89.29 | **89.60** | **84.12** | 86.70 | 84.61 | 81.00 | **67.51** |

Table 1: **Reliability performance on three DST datasets.** *Method denotation*: DPO Joint: SFT and DPO Joint Tranining. DPO Post: DPO Post Training. *Metric denotation*: Prec$_{sure}$: precision of "sure" predictions. Rec$_{total}$: recall of the "sure" and "unsure" predictions. Opti. F1: Optimal-F1. Qual. F1: Quality-F1. Notably, all outputs of the Direct baseline are treated as "sure".

## 4.1 Experiments Setup

### 4.1.1 Model and Baselines

We utilize the LLaMA3 8B model and the Qwen2.5 7B model as the backbone for our experiments. For reliability alignment, we conducted **Reliability-SFT** and integrated DPO to create two additional variants: **Reliability-SFT+DPO Joint Training**, which undergoes joint training with SFT and DPO, and **Reliability-SFT+DPO Post Training**, where DPO training follows initial SFT training, optimizing the model beyond the Reliability-SFT foundation. The key distinction lies in the different source distributions of preference data sampled for DPO. Training details could be found in Appendix §A.

To evaluate the reliability of our proposed models, we compare them against six baselines: three prompt-based and three uncertainty-based approaches (see Appendix §E). Notably, our focus is on the model's self-awareness rather than task performance, so we do not include SOTA DST models. By default, we use only 1,000 samples for Reliability-SFT training and 2,000 for DPO, which is significantly less than traditional models.

**Prompt-based** methods encompass three approaches: Direct, Verbose, and Self-Reflection (denoted as SR, Ji et al. (2023a)). The Direct approach generates predictions without confidence estimation, while Verbose produces both predictions and confidence labels simultaneously. SR assesses each prediction through self-evaluative reflection.

**Uncertainty-based** methods include Token Probability (Prob, Manakul et al. (2023)), Self-Consistency (SC, Chen & Mueller (2024)), and P(True) (Kadavath et al., 2022). These methods use different techniques to classify outputs as "sure" or "unsure". Prob calculates average token probabilities; SC measures the frequency of predictions across multiple samples; and P(True) determines the probability of the "True" token during model self-evaluation.

### 4.1.2 Dataset

We evaluate our methods on the Dialogue State Tracking (DST) task in task-oriented dialogue systems, which involves extracting slot-value pairs from multi-turn dialogues based on a predefined ontology (Xu et al., 2024a). We focus on the intricate multi-slot problem, where the model is likely to exhibit varying confidence levels for different parts of responses to the same query. Consequently, we do not adopt traditional QA or mathematical datasets. To ensure a robust evaluation, we utilize three datasets: MultiWOZ-2.4 (Ye et al., 2021), BiTOD (Lin et al., 2021) and SGD (Rastogi et al., 2020). For BiTOD, we adopt the English version, while for SGD, we randomly select 10,000 instances from the testset to mitigate excessive computational overhead. Common evaluation metrics for DST include Joint Goal Accuracy (JGA) and Slot-F1. For JGA, a sample is deemed correct only if all predictions are accurate without omissions, while Slot-F1 offers a slot-level assessment.

## 4.2 Reliability Evaluation

The experimental results on model reliability are summarized in Table 1. We evaluate our reliability-training pipeline against baseline algorithms, with a particular focus on the model's self-awareness capabilities. For **RQ1**, our method significantly outperforms traditional prompt-based and uncertainty-based methods across various datasets, demonstrating a substantial improvement in self-awareness.

Our approach consistently achieves superior Quality-F1 scores, demonstrating improved ability to distinguish predictions as "sure" or "unsure" based on confidence. A high $Precision_{sure}$ indicates reliable "sure" predictions, fostering user trust with minimal need for verification. Simultaneously, a high $Recall_{total}$ ensures the "unsure" category contains a broad range of potentially correct outputs, which forms a solid basis for refinement. Together, these metrics lead to a higher Optimal-F1 score, representing the theoretical maximum post-refinement performance. In essence, our system optimally balances precision and recall, providing both immediate reliability and a strong potential for post-refinement accuracy.

In our approach, DPO-based methods generally outperform SFT-only methods. Specifically, models trained with DPO Joint Training achieve higher Optimal-F1 scores, suggesting a more conservative stance that defers to subsequent refinement. In contrast, those undergoing Post Training yield superior Quality-F1 scores, indicating greater confidence and higher-quality predictions. This discrepancy arises because Joint Training samples from a pre-SFT distribution, which has poorer performance. This results in a dataset with a higher proportion of "unsure" predictions, leading to more cautious model behavior.

Notably, some baselines, such as Self-Consistency, show high $Precision_{sure}$, $Recall_{total}$, and Optimal-F1 scores, suggesting initial superiority. However, this often indicates an overly conservative model that frequently labels predictions as "unsure." This leads to a suboptimal balance between prediction types, as evidenced by a significantly lower Quality-F1 score. In an extreme case, a model could label all predictions as "unsure", maximizing Optimal-F1 but shifting the entire burden to the refinement stage. This approach sacrifices immediate usability and fails to balance reliability and helpfulness.

## 4.3 Refinement for Unsure Prediction

As described in Section §3.3, we trained an automatic refinement model for each dataset. We refined the "unsure" predictions from the initial stage and merged them with the "sure" predictions to generate the final results. Table 2 shows the overall evaluation. For **RQ2**, our method achieves significantly better performance after refinement, with substantial improvements in Slot-F1 and JGA.

Referring to §4.2, despite their higher theoretical performance limit (Optimal-F1), DPO Joint Training models exhibit suboptimal Slot-F1 and JGA after refinement compared to Post Training models. This discrepancy stems from the limitations of the refinement model. Since the refinement model is not perfect, an excessive number of "unsure" predictions can lead to residual errors after refinement, incurring significant costs without proportional gains. This highlights the importance of a balanced "sure" and "unsure" classification. An ideal model should minimize unnecessary "unsure" predictions and refine only when truly warranted, maximizing both efficiency and overall performance.

| Method Type | Method | MultiWOZ-2.4 | | BiTOD | | SGD | |
|---|---|---|---|---|---|---|---|
| | | Slot-F1 ↑ | JGA ↑ | Slot-F1 ↑ | JGA ↑ | Slot-F1 ↑ | JGA ↑ |
| | | *LLAMA3-8B* | | | | | |
| Prompt | Direct | 78.86 | 36.01 | 82.40 | 36.46 | 32.41 | 13.58 |
| | Verbose | 76.80 | 36.69 | 76.93 | 27.99 | 23.11 | 9.00 |
| | SR | 78.68 | 29.72 | 76.50 | 30.74 | 29.90 | 11.25 |
| Uncertainty | Prob | 75.89 | 24.19 | 73.88 | 24.86 | 28.48 | 9.63 |
| | SC | 70.98 | 20.70 | 69.98 | 24.86 | 25.60 | 4.39 |
| | P(True) | 73.88 | 19.18 | 67.07 | 19.03 | 26.29 | 7.34 |
| Reliability (Ours) | SFT | 85.95 | 53.76 | 85.21 | 60.09 | 74.04 | 27.62 |
| | + DPO Joint | 86.47 | 54.31 | 84.53 | 61.37 | **75.47** | **29.18** |
| | + DPO Post | **86.55** | **56.33** | **85.87** | **64.95** | 73.96 | 28.13 |
| | | *Qwen2.5-7B* | | | | | |
| Prompt | Direct | 78.66 | 36.35 | 83.23 | 37.39 | 42.02 | 16.93 |
| | Verbose | 67.59 | 26.45 | 74.37 | 27.38 | 27.79 | 9.19 |
| | SR | 68.88 | 19.09 | 67.18 | 19.15 | 34.77 | 8.78 |
| Uncertainty | Prob | 74.18 | 23.56 | 73.97 | 25.64 | 35.91 | 10.81 |
| | SC | 59.34 | 23.45 | 65.60 | 28.31 | 34.34 | 3.06 |
| | P(True) | 67.52 | 17.05 | 66.25 | 17.99 | 33.66 | 8.07 |
| Reliability (Ours) | SFT | 84.50 | 51.90 | 85.29 | 58.39 | 74.86 | 28.02 |
| | + DPO Joint | 87.06 | 52.41 | 85.44 | 62.31 | 74.85 | 27.06 |
| | + DPO Post | **87.57** | **53.19** | **85.86** | **63.61** | **74.95** | **28.51** |

Table 2: **Comparison of different methods on overall performance after refinement.** Note: for Direct baseline we conduct a fully refinement since there's no confidence label.

### 4.4 Scalability Analysis

To evaluate scalability, we experimented with foundation models of varying capabilities. We consider the original LLAMA3 8B as a low-performance baseline (denoted as Low), then fine-tuned two models with 1,000 (Medium) and 10,000 (High) samples to enhance their DST capabilities. We then continually conducted reliability training and evaluated the performance of the obtained reliable LLMs against baseline methods. As shown in Figure 3, our methods consistently outperform baselines after refinement, demonstrating strong scalability. This is particularly relevant for real-world scenarios where models are often domain-specifically tuned and possess high task capabilities. Detailed results in Appendix §F further support these findings.

As illustrated in Table 7 in Appendix §F, the advantage of DPO Post Training over Joint Training becomes more pronounced as foundation model performance improves. This is because our offline DPO algorithm's limitations are exacerbated by higher-performing foundation models, negatively affecting the Joint Training data distribution and overall performance. Notably, in the SGD dataset, the benefits of DPO Joint Training diminish with increasing foundation model performance, even falling behind the SFT-only approach.

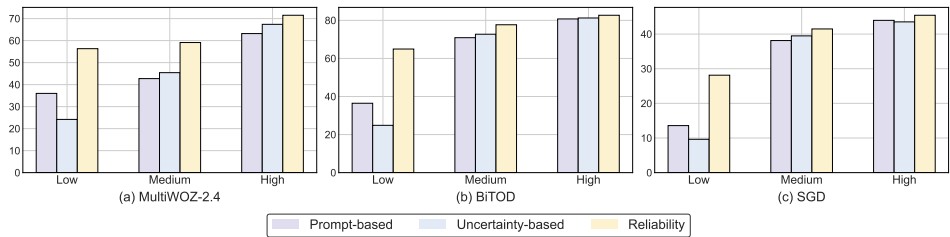

Figure 3: **Comparison of different methods under foundation models of various task ability.** Figure (a), (b) and (c) illustrates the *JGA after refinement* (%) on three datasets. Chosen representative methods: Prompt-based: Direct; Uncertainty-based: Prob; Reliability: SFT+DPO Post Training.

### 4.5 DPO Preference Strategies

We constructed our DPO preference data using Weighted-F1 as the ranking metric, as detailed in §2.2.1. Weighted-F1 incorporates two adjustable $\alpha$ parameters: $\alpha_1$ for Weighted Precision and $\alpha_2$ for Weighted Recall. Using the Reliability-SFT trained LLAMA3 8B model, we generated DPO preference data with varying $\alpha$ values and performed subsequent Post Training. Results on the MultiWOZ-2.4 dataset (Figure 5) show that the model achieves optimal performance at $\alpha_1 = 0.25$ and $\alpha_2 = 0.75$, reaching near-best results at a relatively low cost.

Analysis indicates that increasing $\alpha_1$ heightens sensitivity to errors in "unsure" predictions, reducing tolerance and the number of "unsure" outputs. Conversely, increasing $\alpha_2$ amplifies rewards for successfully recalling "unsure" predictions, encouraging a more conservative stance. The optimal alpha settings depend on the specific scenario and refinement model performance; if the refinement model underperforms, the cost of excessive "unsure" predictions may not be justified. The flexible adjustment of both $\alpha$ parameters allows for adaptation to diverse real-world scenarios.

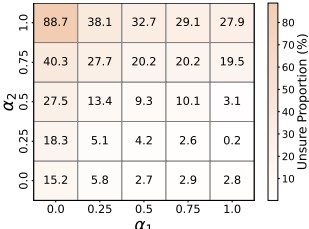

(a) Unsure Proportion

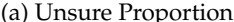
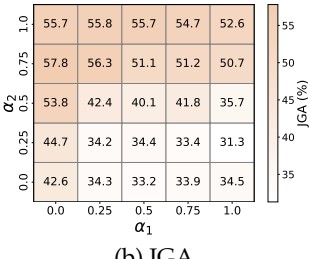

(b) JGA

Figure 5: **Comparison of DPO preference strategies on model behavior and performance.**

### 4.6 Detailed Analysis

#### 4.6.1 Prediction Types Analysis

To evaluate our model's ability to make reliable confidence judgments, we further analyzed the distribution of its output categories and compared them to baselines. We examined five output categories: correct and incorrect predictions classified as "sure" or "unsure", along with missing predictions. The results are presented in Figure 6.

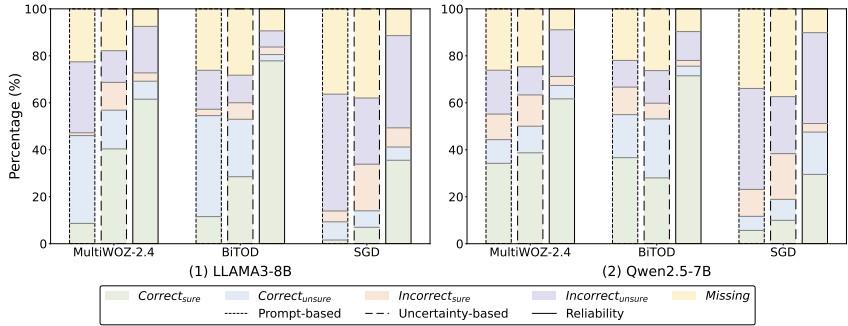

Figure 6: **Percentage of predictions types among different methods.** Chosen representative methods: Prompt-based: Direct; Uncertainty-based: Prob; Reliability: SFT+DPO Post Training.

Our findings highlight two key improvements: (1) **Enhanced reliability**: We observed a notable reduction in incorrect "sure" predictions, with most errors concentrated in the "unsure" category. This indicates a stronger self-assessment capability, allowing the model to better identify potential errors than baselines. Additionally, the "unsure" category consistently contains fewer correct predictions than incorrect ones, a distinction baseline models struggle to achieve. (2) **Improved recall**: Our approach significantly reduces missing predictions by capturing previously omitted relevant information within the "unsure" category. While some of these may still contain errors, this shift enhances overall helpfulness and provides a stronger foundation for refinement.

### 4.6.2 Cost Analysis

In this work, **Cost** refers to the additional overhead from the refinement process. We define the cost of refining an "unsure" prediction as a constant value of 1, equating Cost to the number of "unsure" predictions. For better comparison, we express Cost as the proportion of "unsure" predictions.

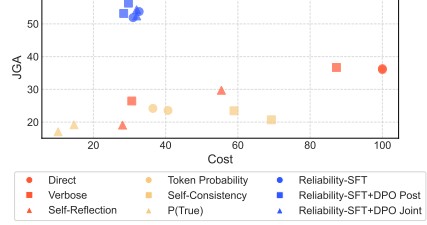

Figure 7: **Comparison of different methods of Cost-JGA on MultiWOZ-2.4.**

We compare the performance and Cost of various baselines in Figure 7. Results are shown for the MultiWOZ-2.4 dataset, with complete findings in Appendix §G. Our methods show significant performance improvements at a relatively low Cost. Additionally, as shown in Appendix Table 8, Cost decreases significantly as the foundation model's performance improves, which is expected as a stronger model produces fewer uncertain predictions.

### 4.6.3 Refinement Model Comparison

We used various refinement models to enhance predictions from the LLAMA3 Reliability-SFT+DPO Post model. As shown in Table 8, we compared our trained models against GPT-4o and the DeepSeek series (DeepSeek-AI et al., 2025) (8B and 70B), all known for strong reasoning. Results show significant performance differences, with DeepSeek 8B underperforming while the 70B model nearly matches GPT-4o. Our task-specific fine-tuning with CoT substantially improves refinement accuracy.

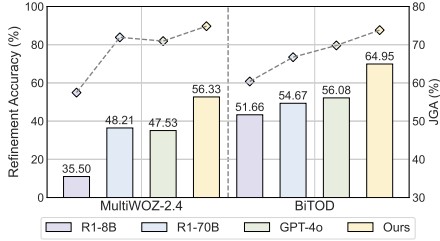

Figure 8: **Comparison of different refinement models.** The line chart represents "Refinement Accuracy", while the histogram represents "JGA".

### 4.7 Extension to Mathematical Problems

To validate the generalization of our method to different tasks, we conducted supplementary experiments on the widely used **GSM8K** mathematics dataset (Cobbe et al., 2021) using the Qwen2.5-7B-Instruct model with 1,000 training samples. We evaluated the model under two test settings: (a) with Chain-of-Thought (CoT) reasoning (Wei et al., 2023) and (b) by directly predicting the final answer. We tested both settings as LLMs generally perform

differently on each. For the refinement model, we used the DeepSeek-R1-Distill-Llama-70B model, which has strong mathematical reasoning capabilities.

We adapted the F1-based evaluation metric from Section §2.2 to an accuracy-based definition:

$$\text{Weighted-Accuracy}(X, \alpha) = \frac{\sum\limits_{x_i \in Sure} S(x_i) + \alpha \times \sum\limits_{x_j \in Unsure} S(x_j)}{N_{Sure} + \alpha \times N_{Unsure}}$$

$$\text{Quality-Accuracy}(X) = \text{Weighted-Accuracy}(X, 0.5)$$

As shown in Table 3, our method consistently outperformed both prompt-based and uncertainty-based baselines in improving the model's self-assessment ability, achieving higher Quality-Accuracy. Furthermore, after applying refinement with the same refinement model, our approach yielded a higher final accuracy at a reasonable computational cost.

| Method Type | Method | Quality Accuracy | Final Accuracy |
|---|---|---|---|
| Prompt | Direct | 86.81 | 86.81 |
| | Verbose | 90.8 | 90.98 |
| Uncertainty | Prob | 87.69 | 88.55 |
| | SC | 89.41 | 91.48 |
| Reliability (ours) | SFT | 89.69 | 91.05 |
| | + DPO Joint | 90.14 | 93.10 |
| | + DPO Post | 92.73 | 95.30 |

(a) Inference setup: Chain-of-Thought

| Method Type | Method | Quality Accuracy | Final Accuracy |
|---|---|---|---|
| Prompt | Direct | 19.94 | 19.94 |
| | Verbose | 21.44 | 24.34 |
| Uncertainty | Prob | 24.49 | 76.50 |
| | SC | 24.37 | 70.05 |
| Reliability (ours) | SFT | 27.31 | 77.94 |
| | + DPO Joint | 26.16 | 78.24 |
| | + DPO Post | 27.83 | 79.83 |

(b) Inference setup: Answer Directly

Table 3: **Comparison of different methods on GSM8K dataset.** *Quality Accuracy* assigns a weight of 0.5 to unsure samples. *Final Accuracy* represents the overall accuracy after refinement.

## 5 Related Work

The knowledge boundary of LLMs has emerged as a critical topic in recent research, high-lighting their limitations in generating reliable outputs (Yin et al., 2024; Li et al., 2024b). Misalignment between model behavior and knowledge boundaries can result in factual hallucinations (Huang et al., 2023) and ambiguous responses (Liu et al., 2023).

Various methods have been proposed to assess these boundaries. Uncertainty-based approaches quantify prediction confidence through token probabilities (Manakul et al., 2023) and consistency (Chen & Mueller, 2024). Calibration strategies, including prompting (Tian et al., 2023) and fine-tuning (Tao et al., 2024) for improved confidence expression, align model confidence with prediction accuracy. Internal State Probing evaluates prediction factuality by analyzing model states, such as activations (Li et al., 2024a). While effective, these methods often rely on external post-hoc techniques, which are computationally expensive and lack integration with the model's endogenous reasoning processes (Huang et al., 2024; Zhou et al., 2024; Pan et al., 2025; Shan et al., 2025).

To further align model behavior rather than merely estimate uncertainty, some studies guide models in self-reflection to minimize self-contradictions (Wei et al., 2022; Ji et al., 2023b), albeit with increased computational overhead. More recently, RL-based approaches leverage uncertainty estimation to train reward models, enhancing LLM truthfulness by rejecting queries beyond their capabilities (Xu et al., 2024b; Chen et al., 2024; Xue et al., 2024). While effective, these methods rely on the accuracy of uncertainty estimation algorithms and meanwhile may unintentionally reduce overall helpfulness.

## 6 Conclusion

In this work, we propose a two-stage framework that enhances the reliability of large language models by integrating fast and slow thinking paradigms. Through precise confidence assessment and high-accuracy refinement, we achieve a balance between reliability and usability. Extensive evaluations demonstrate that our method significantly improves model self-awareness and performance across complex tasks, establishing a new paradigm for enhancing the reliability of language models in error-sensitive applications.

## Acknowledgments

This work is funded by the China NSFC Projects (62120106006, 92370206, and U23B2057) and Shanghai Municipal Science and Technology Projects (2021SHZDZX0102 and 25X010202846).

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

## A Training Details

We modify the DeepSpeed-Chat (Yao et al., 2023) framework to accommodate our experimental setup. We adopt most of the default training parameters provided by DeepSpeed-Chat except that the learning rate for DPO training is set to 1e-7 and the maximum sequence length is set to 8096 on the SGD dataset and 2048 on the others. All models are trained for 1 epoch using NVIDIA A800 GPUs.

## B Execution Example for Two Stages

We provide an example sample that includes the complete execution flow and model outputs for the two stages as shown in Table 4.

| **Input** |
| --- |
| Dialog history:
USER: Hi, I'm looking for a hotel to stay in.
....
USER: Okay, please book that for 3 people and 2 nights starting from Friday.
SYSTEM: Booking was successful. Reference number is : 9HMD04UW. anything else?
USER: I would love to find a restaurant as well.
...

Reliable dialog state: |
| **Stage One: Reliable LLM Predictions** |
| {"hotel": {"bookday": {"value": "friday", "confidence": "sure"}, "bookpeople": {"value": "3", "confidence": "sure"}, "bookstay": {"value": "2", "confidence": "sure"}}, "restaurant": {"bookpeople": {"value": "3", "confidence": "unsure"}}} |
| **Stage Two: Refinement for "Unsure" Predictions:** |
| **Target:** {"domain": "restaurant", "slot": "bookpeople", "value": "3"}
**Refinement model:**
Thinking steps:
step1: structure aspect: "bookpeople" is one of the slots in the restaurant domain and the value is specified as "3".
step2: semantic aspect: In this case, the user never specifies the number of people for the restaurant booking. The number "3" is related to the hotel booking, not the restaurant. Therefore, the slot-value pair is irrelevant to the user's intent for the restaurant and should be removed.
step3: the predicted slot-value pair is incorrect in semantic, and the slot should be removed.
Refine action: removing
Refined slot-value pair: none |
| **Final Refined Predictions:** |
| {"hotel": {"bookday": {"value": "friday", "confidence": "sure"}, "bookpeople": {"value": "3", "confidence": "sure"}, "bookstay": {"value": "2", "confidence": "sure"}}} |

Table 4: **An example of the complete execution flow and model outputs.**

## C Training Data Construction

We provide detailed examples to illustrate our training data construction process, including model inputs and the expected outputs the model is intended to learn. The data construction for SFT is outlined in Table 5, while that for DPO is detailed in Table 6.

| Input |
|---|
| Dialog history:
USER: Is there a modern European restaurant in the north?
SYSTEM: I found city stop restaurant. They are located at Cambridge City Football Club Milton Road Chesterton.
USER: That sounds lovely!
SYSTEM: Would you like me to help you with reserving a table?
USER: Maybe. Is that restaurant in the moderate price range?
SYSTEM: No, it is in the expensive price range. Is this okay, or would you like to try another area?
USER: How about a restaurant that serves Indian food?
SYSTEM: There are several restaurants what part of town would you like it?
USER: A restaurant in the north, please.

Reliable dialog state: |

| **Gold Labels** |
|---|
| {"restaurant": {"area": "north", "food": "indian", "pricerange": "moderate"}} |

| **LLM Original Predictions** |
|---|
| {"restaurant": {"area": "north", "food": "european", "name": "Cambridge City Football Club Milton Road Chesterton"}} |

| **Constructed SFT Sample Output** |
|---|
| {"restaurant": {"area": {"value": "north", "confidence": "sure"}, "food": {"value": "European", "confidence": "unsure"}, "pricerange": {"value": "moderate", "confidence": "unsure"}, "name": {"value": "Cambridge City Football Club Milton Road Chesterton", "confidence": "unsure"}}} |

| **Explanation** |
|---|
| In the LLM's original predictions, the value for *area* is correct, *food* is incorrect, *name* is extraneous, and *pricerange* is missing. Consequently, in the constructed SFT sample output, *area* is labeled as "sure," while the other three are marked as "unsure," with erroneous information left uncorrected. |

Table 5: **An example of the SFT data construction.**

# D   Refinement Model

We trained an automated refinement model for each dataset. The refinement model conducts in-depth step-by-step reasoning on unsure predictions based on task-specific criteria, operating within a defined action space. Our refinement models are built upon LLAMA3 8B utilizing SFT training with data constructed using OpenAI's GPT-4o.

**Action Space**   The action space of the refinement model includes three actions: remove, reserve, and correct. A prediction should be reserved if entirely correct, should be corrected if partially correct and should be removed if entirely unnecessary. For any unsure predictions, a proper action from these three options is sufficient to achieve the required refinement.

**Data Construction**   We adopt a multi-step reasoning Chain-of-Thought (CoT) paradigm. The specific reasoning instructions and processes are detailed in Table 10. For data construction, we used OpenAI's GPT-4o model. During the data generation process, we provided the model with the prediction to be refined and the corresponding golden label, then instructed it to generate a reasoning process based on predefined principles, as shown in Table 11. We combined the instruction, input, the prediction to be refined, the thinking steps, and the refinement result into a final training sample. For each dataset, we sampled 5,000 examples

| **Input** |
| --- |
| Dialog history: 
 USER: Is there a modern European restaurant in the north? 
 SYSTEM: I found city stop restaurant. They are located at Cambridge City Football Club Milton Road Chesterton. 
 USER: That sounds lovely! 
 SYSTEM: Would you like me to help you with reserving a table? 
 USER: Maybe. Is that restaurant in the moderate price range? 
 SYSTEM: No, it is in the expensive price range. Is this okay, or would you like to try another area? 
 USER: How about a restaurant that serves Indian food? 
 SYSTEM: There are several restaurants what part of town would you like it? 
 USER: A restaurant in the north, please. 

 Reliable dialog state: |
| **Gold Labels** |
| {"restaurant": {"area": "north", "food": "indian", "pricerange": "moderate"}} |
| **Multi-Sampling With the Foundation LLM** |
| $Sample_1$ 
 ... 
 $Sample_n$ |
| **Constructed DPO Sample Output Pairs** |
| **Chosen** 
 {"restaurant": {"area": {"value": "north", "confidence": "sure"}, "food": {"value": "indian", "confidence": "sure"}, "pricerange": {"value": "moderate", "confidence": "unsure"}}} 
 **Rejected** 
 {"restaurant": {"area": {"value": "north", "confidence": "sure"}, "food": {"value": "indian", "confidence": "sure"}}} |
| **Explanation** |
| Compared to the rejected sample, the chosen sample exhibits a higher recall for a correct *pricerange* within the "unsure" category, aligning more closely with our alignment objectives. This is reflected in a higher Weighted-F1 score, justifying its selection over the rejected sample. |

Table 6: **An example of the DPO data construction.**

from the model's unsure predictions as well as from randomly selected data in the training set, ensuring a balanced distribution of data across three actions.

# E   Baselines Details

Here we detail the specifics of the baselines.

**Prompt-based**

- Direct: Utilizes prompts similar to those in Table 13 to guide the model in generating predictions directly, without incorporating confidence labels.

- Verbose: Employs prompts akin to those in Figure 13 to direct the model, which outputs confidence labels (sure or unsure) alongside predictions based on its self-assessment.

- Self-Reflection (SR): Builds on predictions from the Direct baseline, allowing the model to reflect on each prediction to assess its confidence level.

**Uncertainty-based**

- Token Probability (Prob): Computes average token-level probabilities, classifying outputs as "sure" or "unsure" based on a threshold $t$, with probabilities normalized and $t$ set to 0.8.
- Self-Consistency (SC): Samples $N$ times, classifying predictions that appear consistently at proportion $p$ as "sure," while others are marked as "unsure." We conduct ten samples ($N = 10$) and set $p$ to 50%.
- P(True): Based on predictions from the Direct baseline, this method evaluates the correctness of each prediction individually, marking those with a higher probability of responding "True" than "False" as "sure," otherwise as "unsure."

# F Scalability Analysis

The detailed results are presented in Table 7, demonstrating that our method maintains superiority across various capability baselines, achieving more reliable outcomes (higher Precision$_{sure}$, Recall$_{total}$, and Quality-F1) and enhanced overall performance post-refinement (higher JGA).

| Method Type | Method | MultiWOZ-2.4 | | | | BiTOD | | | | SGD | | | |
|---|---|---|---|---|---|---|---|---|---|---|---|---|---|
| | | Prec$_{sure}$ ↑ | Rec$_{total}$ ↑ | Qual. F1 ↑ | JGA ↑ | Prec$_{sure}$ ↑ | Rec$_{total}$ ↑ | Qual. F1 ↑ | JGA ↑ | Prec$_{sure}$ ↑ | Rec$_{total}$ ↑ | Qual. F1 ↑ | JGA ↑ |
| | | *1000Samples-Trained LLAMA3 Based* | | | | | | | | | | | |
| Prompt | Direct | 91.07 | 86.28 | 87.84 | 42.73 | 93.01 | 92.78 | 92.82 | 70.91 | 82.28 | 85.10 | 82.77 | 38.13 |
| Uncertainty | Prob | 91.84 | 86.52 | 86.84 | 45.43 | 94.93 | 93.55 | 93.66 | 72.73 | 84.45 | 85.24 | 82.80 | 39.48 |
| | SC | 92.68 | **92.37** | 84.31 | 43.18 | 95.70 | 95.12 | 92.05 | 70.37 | 86.61 | 92.30 | 78.71 | 28.62 |
| Reliability (Ours) | SFT | 95.42 | 92.11 | 89.86 | 55.62 | 94.97 | 94.74 | 93.80 | 73.64 | 85.16 | 85.14 | 81.06 | 39.23 |
| | + DPO Joint | 95.11 | 90.68 | 89.05 | 58.94 | 96.06 | **95.60** | 95.11 | 75.36 | 81.66 | 82.82 | 79.69 | 37.62 |
| | + DPO Post | **96.33** | 92.15 | **89.91** | **59.12** | **96.22** | 95.54 | **95.43** | **77.70** | **86.67** | **85.44** | **83.27** | **41.48** |
| | | *10000Samples-Trained LLAMA3 Based* | | | | | | | | | | | |
| Prompt | Direct | 94.51 | 93.30 | 93.63 | 63.19 | 95.35 | 95.14 | 94.34 | 80.76 | 83.24 | 83.75 | 83.11 | 43.96 |
| Uncertainty | Prob | 95.77 | 94.06 | 92.95 | 67.39 | 95.95 | 95.53 | 95.23 | 81.26 | 84.37 | 84.61 | 83.08 | 43.52 |
| | SC | 96.24 | **96.40** | 92.13 | 67.61 | **97.83** | **97.81** | 93.42 | 78.04 | **88.13** | **92.42** | 79.26 | 38.20 |
| Reliability (Ours) | SFT | 96.98 | 95.86 | 94.40 | 70.85 | 96.44 | 96.20 | 96.03 | 80.54 | 85.30 | 83.92 | 83.57 | 45.11 |
| | + DPO Joint | 96.98 | 95.61 | 94.71 | 70.84 | 97.15 | 96.65 | 95.96 | 82.35 | 82.31 | 82.36 | 80.75 | 40.48 |
| | + DPO Post | **97.71** | 96.32 | **94.80** | **71.51** | 96.57 | 96.35 | **96.59** | **82.69** | 86.29 | 85.53 | **84.21** | **45.43** |

Table 7: **Reliability performance of methods on models with higher task capability.** Note: the JGA column indicates results after refinement.

# G Refinement Cost

The detailed overall results of cost are presented in Table 8 and 9. The relationships between cost and performance for the BiTOD and SGD datasets are illustrated in Figure 9a and Figure 9b, respectively, where *cost* refers to the proportion of unsure predictions.

| Method Type | Method | MultiWOZ-2.4 | | BiTOD | | SGD | |
|---|---|---|---|---|---|---|---|
| | | LLAMA | Qwen | LLAMA | Qwen | LLAMA | Qwen |
| Prompt | Direct | 100.00 | 100.00 | 100.00 | 100.00 | 100.00 | 100.00 |
| | Verbose | 87.28 | 30.62 | 80.69 | 38.05 | 90.30 | 74.07 |
| | SR | 55.43 | 28.07 | 95.50 | 8.55 | 84.77 | 15.07 |
| Uncertainty | Prob | 36.47 | 40.65 | 50.46 | 52.91 | 56.68 | 53.01 |
| | SC | 69.26 | 58.91 | 58.05 | 62.59 | 96.80 | 92.71 |
| | P(True) | 14.63 | 10.22 | 13.35 | 1.82 | 1.52 | 1.71 |
| Reliability (Ours) | SFT | 32.67 | 30.98 | 20.03 | 21.78 | 51.24 | 55.96 |
| | + DPO Joint | 31.96 | 31.98 | 21.19 | 23.63 | 57.72 | 45.29 |
| | + DPO Post | 27.69 | 28.38 | 10.54 | 18.14 | 40.68 | 53.10 |

Table 8: **Cost comparison of different methods using the original LLAMA3-8B and Qwen2.5-7B models.**

| Method Type | Method | MultiWOZ-2.4 | | BiTOD | | SGD | |
|---|---|---|---|---|---|---|---|
| | | Medium | High | Medium | High | Medium | High |
| Prompt | Direct | 100.00 | 100.00 | 100.00 | 100.00 | 100.00 | 100.00 |
| Uncertainty | Prob | 6.55 | 5.74 | 3.29 | 1.37 | 8.14 | 5.53 |
| | SC | 28.93 | 12.17 | 6.25 | 2.12 | 41.68 | 41.42 |
| Reliability (Ours) | SFT | 13.52 | 5.12 | 2.16 | 0.66 | 12.60 | 1.87 |
| | + DPO Joint | 10.70 | 3.94 | 1.36 | 1.04 | 11.66 | 2.67 |
| | + DPO Post | 13.32 | 8.59 | 3.44 | 1.46 | 19.01 | 4.19 |

Table 9: **Cost comparison of reliability-trained models using different foundation models**: *Medium* denotes a model that has acquired certain task capabilities through SFT with 1,000 task-specific samples, while *High* refers to the 10,000-samples-trained one.

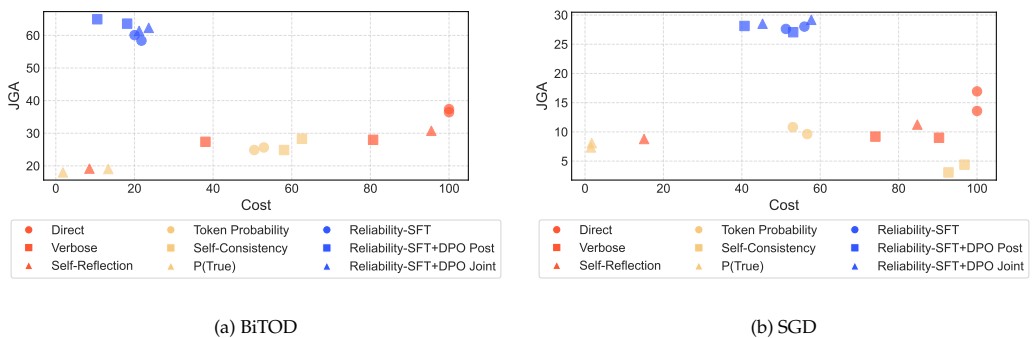

(a) BiTOD                                          (b) SGD

Figure 9: **Comparison of different methods in terms of performance and cost.**

---

**Refinement Examples**

Give you a dialog history between USER and SYSTEM, and a slot-value pair extracted from the dialog, I want you to analyze on the dialog history and review/refine the given dialog state slot-value pair to make them more accurate and reliable.
**## Domain Slot Spaces**
{
    "hotel": {
        "name": {
            "data_type": str,
            "example": "hamilton lodge"
        },
        ...
    },
    ...
}
**## Question**
**Dialog history**:
USER: Hello. I need train to London liverpool Street. SYSTEM: Where are you departing from and do you have a time preference? USER: Yes, I'd like to leave Cambridge sometime after 7:00. SYSTEM: I have one that leaves at 7:59 and 4 more that depart every two hours after. USER: The 7:59 will be fine.
**Predicted slot-value pair**:
train: leaveat = 7:59
**Refinement**:
Thinking steps:
step1: structure aspect: "leaveat" is one of the slots in the train domain and the value is incorrectly specified as "7:59" (formation error). Precise time should strictly follow the "HH:MM" format.
step2: semantic aspect: "leaveat" is not a "name"-related slot, so the value must be extracted from the user's direct intent. Although the system provides a train that departs at 7:59, the user only mentions that he'd like to leave Cambridge after 7:00. Considering the formation, the correct value should be 07:00.
step3: domain train correct; slot leaveat correct; value 7:59 incorrect.
step4: the predicted slot-value pair is incorrect in both structure and semantic, and the value should be corrected.
**Refine action**: correct
**Refined slot-value pair**: train: leaveat = 07:00

Table 10: **An example of the Refinement process.**

---

**GPT4o Prompt For Refinement Thinking Steps Generation**

---

Give you a dialog history between USER and SYSTEM, a predicted slot-value pair and a gold refinement, I want you to analyze on the dialog history, refer to the examples I've provided and the golden refinement, and then continue to provide the thinking steps following the predefined principles.

Firstly, we use json dict to describe the slots and their corresponding value space. Then, we will specify the requirements you need to comply and provide some examples. Last, we will present you with the dialog history, the predicted dialog state slot-value pair and the gold refinement for you to perform continuation.

## Domain Slot Spaces

```
{
    "hotel": {
        "name": {
            "data_type": str,
            "example": "hamilton lodge"
        },
        ...
    },
    ...
}
```

## Meta Requirements

1. Analyze the dialog history and the predicted slot-value pair carefully, correct and refine the pair according to the following Detailed Refinement Principles.

2. For the given slot-value pair, you should only take one of the actions of "reserve", "remove" or "correct" according to the following principles.

   a. reserve: if the slot is relevant to the dialog history and the value is correctly specified, you should keep the pair.

   b. remove: if the slot is irrelevant to the given dialog history or there's not corresponding value could be extracted from the history, you should print "none" indicating that the pair has been removed.

   c. correct: if the slot is relevant to the dialog history and the value is incorrectly specified, you should extract the correct value and replace it.

3. You need to think and reason in the same way as the Examples. Please refer to the Examples for formatting requirement. Make sure the output is all lowercase. Only focus on the given slot-value pair and do not consider any other slot. only provide the reasonable thinking steps, do not provide any extra prefixes or suffixes.

## Detailed Refinement Principles

· · ·

## Examples

· · ·

## Question

**Dialog history**:

USER: ... SYSTEM: ...

**Predicted slot-value pair**:

train: leaveat = 7:59

**Refinement**:

**Refine action**: correct

**Refined slot-value pair**: train: leaveat = 07:00

Thinking steps:

---

Table 11: **The prompt used to instruct GPT-4o to generate thinking steps for refinement.**

# H   LLM Instructions

The instructions used in this work are presented in this chapter, taking the MultiWOZ-2.4 dataset as an example. The instructions for the BiTOD dataset are similar.

### H.1 Inference Instruction

The prompts used with the original untrained LLAMA3 model are shown in Tables 12 and 13. Among them, the Direct Inference Instruction requires the model to directly output the prediction results, while the Reliability Inference Instruction additionally requires the model to output a confidence label alongside the prediction.

---

**Direct Inference Instruction**

---

Give you a dialog history between USER and SYSTEM, I want you to analyze on it and generate the dialog state.
Firstly, we use json dict to describe the slots and their corresponding value space in each domain. Then, we will specify the requirements you need to comply. Last, we demonstrate some use cases.
## Domain and Slot Space
The availabel ontology of the dialog state is as follows:
```
{
    "hotel": {
        "name": {
            "data_type": str,
            "example": "hamilton lodge"
        },
        ...
    },
    ...
}
```
## Requirements
1. Analyze the dialog history carefully and fill the relevant domains and slots.
2. For slots with a specified value range, responses must fall within the provided range. For slots without specified value range, the answer must be extracted from the history. Set value as "dontcare" if user doesn't have a preference.
3. You only need to consider the domains and slots that are relevant to the conversation history. Do not include those irrelevant in your response and avoid presenting empty domains or slots.
4. Your answer should also be in one-line jsonl format and make sure the output is all lowercase. Do not provide any extra prefixes or suffixes or any explanations.
## Output Dialog State Example
{"hotel": {"area": "centre", "name": "alexander bed and breakfast", "parking": "yes", "type": "guesthouse"}, "attraction": {"name": "kambar"}}
## Examples
shots...

---

Table 12: **Instruction used in Direct, Token probability and Self-consistency baselines.**

### H.2 Training Instruction

The prompts used with the trained LLAMA3 model are shown in Tables 14 and 15. The Direct Training Instruction is designed to train foundation models capable of directly outputting prediction results. In contrast, the Reliability Training Instruction is designed to train models capable of explicitly outputting confidence labels alongside predictions.

---

**Reliability Inference Instruction**

---

Give you a dialog history between USER and SYSTEM, I want you to analyze on it and generate the dialog state.

Firstly, we use json dict to describe the slots and their corresponding value space in each domain. Then, we will specify the requirements you need to comply. Last, we demonstrate some use cases.

## Domain and Slot Space

The availabel ontology of the dialog state is as follows:

```
{
     "hotel": {
          "name": {
               "data_type": str,
               "example": "hamilton lodge"
          },
          ...
     },
     ...
}
```

## Requirements

1. Analyze the dialog history carefully and fill the relevant domains and slots.

2. For slots with a specified value range, responses must fall within the provided range. For slots without specified value range, the answer must be extracted from the history. Set value as "dontcare" if user doesn't have a preference.

3. You should try to cover as many domians and slot-value pairs relevant to the conversation history as possible. Mark each slot with either "sure" or "unsure" according to your confidence in it. For slot-value pairs that you are pretty sure that they are truly relevant and should be included, and meanwhile you have great confidence in the correctness of the value, you should tag it with "sure". Otherwise, you should tag the slot-value pair with "unsure".

4. Principles:

   a. Goal one: Achieving as close to 100% accuracy as possible in those "sure" slot-value pairs.

   b. Goal two: The "sure" part along with the "unsure" part should cover all possible slots involved in the dialog history as much as possible.

   c. Heavy penalization: Providing incorrect slot-value pairs in the "sure" part.

   d. Heavy penalization: Missing slot-value pairs that should be extracted.

   e. Light penalization: Providing incorrect or redundant slot-value pairs in the "unsure" part.

5. Your answer should also be in one-line jsonl format and make sure the output is all lowercase. Do not provide any extra prefixes or suffixes or any explanations. Please refer to the Output Dialog State Example for formatting requirement.

## Output Dialog State Example

{"hotel": {"area": {"value": "centre", "confidence": "sure"}, "name": {"value": "alexander bed and breakfast", "confidence": "sure"}, "parking": {"value": "yes", "confidence": "unsure"}, "type": {"value": "guesthouse", "confidence": "sure"}}, "attraction": {"name": {"value": "kambar", "confidence": "unsure"}}}

## Examples

shots...

---

Table 13: **Instruction used with original LLAMA model in Verbose baselines.**

---

**Direct Training Instruction**

---

Generate the dialogue state based on the given dialogue context.

---

Table 14: **Instruction used in Direct SFT Training.**

**Reliability Training Instruction**

Generate the dialogue state based on the given dialogue context. Ensure the results are as reliable as possible, the 'sure' parts are as accurate as possible, and the overall coverage includes all relevant slots.

Table 15: **Instruction used in Reliable Training, guiding the model to explicitly extinguish between "sure" and "unsure" responses.**

