# OpenReview forum: "Enhancing LLM Reliability via Explicit Knowledge Boundary Modeling"
_colmweb.org/COLM/2025/Conference — COLM 2025_

### Official Review · Reviewer_gFAY · 2025-05-07

**Rating:** 7
**Confidence:** 4
**Ethics Flag:** 1

**Summary:**

The authors propose a method to increase the generation reliability of LLMs by first training the model via SFT and DPO to produce confidence-labeled responses (using a binary schema of "sure" and "unsure") and then refining unsure responses in a second step with a separate refinement model. This approach explicitly models the inherent trade-off between reliability (producing correct responses) and helpfulness (producing responses as often as possible instead of deflecting) and the authors present various experiments illustrating this trade-off. The method outperforms existing methods for uncertainty-based rejection on three different dialog state tracking datasets.

**Questions To Authors:**

Please comment on the points mentioned under reasons to reject

**Reasons To Accept:**

The paper is well-written and clearly outlines the motivation and methodology. The experiments are informative and clearly demonstrate the utility of the proposed method. I don't see any flaws in the experimental setup. The method is a valuable contribution to increase the reliability of LLM generations.

**Reasons To Reject:**

The paper could explain more clearly how exactly the methodology is applied to the datasets used in the experiments, the details are in the paper but almost all are buried in the appendix. The paper would also be stronger if the experiments would cover a broader range of tasks (especially given that most parts of the paper present the methodology very generically without references to DST). It further remains unclear if previous work also evaluated in the same setup or if this paper is the first one to apply the selected baselines in this setting.

---

> ### Author Response · Authors · 2025-06-02
>
> Thanks for your detailed review and valuable feedback. We provide our responses below:
>
> # Re. to R1: The experimental details are primarily placed in the appendix.
> We sincerely apologize for this inconvenience. Due to length constraint, we had to place some of the experimental details in the appendix, which might have affected the clarity of the main text. We truly appreciate your suggestion and will address this issue in the revision. Specifically, if we have the opportunity to expand the main paper, we will integrate these important experimental settings more clearly. If space is still limited, we will also consider streamlining the text to better accommodate these key details within the main body.
> # Re. to R2: The experiments focus solely on the DST task.
> Thank you very much for your suggestion. In response, we have also conducted experiments on a classic mathematical dataset, **GSM8K**, applying our alignment method and comparing it with the baseline methods mentioned in our paper. The experimental results indicate that our method, compared to the baselines, is highly effective in enhancing the model’s self-awareness ability.
>
> For the GSM8K task, we evaluated the model under two different test settings: (a) using Chain-of-Thought (CoT) reasoning and (b) directly predicting the final answer. LLMs generally perform differently under these two settings, so we tested both. We conducted the experiments on the Qwen2.5-7B-Instruct model and used 1000 samples as training set. For the refinement model, we used the DeepSeek-R1-Distill-Llama-70B model, which has strong mathematical reasoning capabilities.
>
> The results are summarized in the accompanying table, where the metric is adapted from the paper’s F1-based definitions to accuracy-based definitions:
>
> $$
> Weighted\ Accuracy(X,\alpha)\ = \frac{\sum\limits_{x_i\in Sure}S(x_i) + \alpha \times \sum\limits_{x_j\in Unsure}S(x_j)}{N_{Sure}+\alpha\times N_{Unsure}}
> $$
>
> $$
> Quality\ Accuracy(X)\ = Weighted\ Accuracy(X, 0.5)
> $$
>
> Here, the value of $S(x)$ is either 0 or 1, representing whether x is correct. The value of $\alpha$ is in the interval [0, 1].
>
> - **Quality_accuracy** assigns a weight of 0.5 to unsure samples.
> - **Final_accuracy** represents the model’s overall accuracy after refinement of "unsure" predictions.
>
> Results for experiment: Answer with CoT Reasoning:
>
> | Method Type | Method | Quality accuracy | Final accuracy |
> | --- | --- | --- | --- |
> | Prompt | Direct | 86.81 | 86.81 |
> | Prompt | Verbose | 90.8 | 90.98 |
> | Uncertainty | Prob. | 87.69 | 88.55 |
> | Uncertainty | Self Consistency | 89.41 | 91.48 |
> | Reliability(ours) | SFT | 89.69 | 91.05 |
> | Reliability(ours) | SFT + DPO Joint | 90.14 | 93.1 |
> | Reliability(ours) | SFT + DPO Post | 92.73 | 95.3 |
>
> Results for experiment: Answer without CoT Reasoning:
> | Method Type | Method | Quality accuracy | Final accuracy |
> | --- | --- | --- | --- |
> | Prompt | Direct | 19.94 | 19.94 |
> | Prompt | Verbose | 21.44 | 24.34 |
> | Uncertainty | Prob. | 24.49 | 76.5 |
> | Uncertainty | Self Consistency | 24.37 | 70.05 |
> | Reliability(ours) | SFT | 27.31 | 77.94 |
> | Reliability(ours) | SFT + DPO Joint | 26.16 | 78.24 |
> | Reliability(ours) | SFT + DPO Post | 27.83 | 79.83 |
>
> In both evaluation settings, our alignment method significantly outperformed both prompt-based and uncertainty-based baselines in improving the model’s self-assessment ability—specifically, the ability to accurately label predictions as sure or unsure during inference. Our method achieved higher Quality-Accuracy compared to the baselines, with the accuracy of predictions labeled as certain being markedly higher than that of the unsure-labeled answers. It is important to note that **the model's inherent mathematical performance did not improve**: the accuracy of predictions without labels was nearly identical to that of the original Qwen2.5-7B-Instruct model. Moreover, after applying refinement with the same refine model, our method achieved higher final accuracy at a reasonable cost.
> # Re. to R3: Concerns about the experimental setting.
> We would like to clarify that, to the best of our knowledge, our work is the first to explicitly incorporate sure/unsure labels in task-oriented dialogue settings, propose the associated Optimal F1 and Quality F1 metrics, and **apply the selected baselines within this new setting**.
>
> (1) **For tasks**, many prior alignment-related studies have predominantly adopted mathematical or QA-style tasks, which do not necessarily align with the task-oriented dialogue (DST) scenario that we target.
>
> (2) **For baselines such as uncertainty-based methods**, most previous works treat predictions with relatively low confidence as discardable—rather than viewing them as potentially correct answers that could be further refined by a separate refinement model. As a result, the final evaluation metrics in these works tend to be accuracy-focused, which does not align with our emphasis on assessing the model’s self-awareness performance.

---

> > ### Comment · Reviewer_gFAY · 2025-06-04
> >
> > Thank you. I am satisfied with the author's response and will keep my original rating.

---

> > > ### Author Response · Authors · 2025-06-04
> > >
> > > Thank you very much for the time and effort devoted to evaluating our work and for the thoughtful feedback.

---

### Official Review · Reviewer_xXcS · 2025-05-12

**Rating:** 7
**Confidence:** 5
**Ethics Flag:** 1

**Summary:**

This paper proposes to use a reliability model to improve task oriented dialogue systems. Using such a judge model is not a new idea (e.g.,  Dey et al., Accountability Model) but clearly it is impactful and necessary for any agentic model.

**Reasons To Accept:**

The paper seems to be revised multiple times with extensive experimental results and ablations. As is, I believe it is ready for COLM. Maybe the authors can consider a newer benchmark dataset like Tau-Bench to add.

**Reasons To Reject:**

Most agentic systems have severe latency requirements. Having a judge model may not be feasible. It would have been good if the authors also show latency numbers and solutions for that such as caching.

---

> ### Author Response · Authors · 2025-06-02
>
> Thanks for your detailed review and valuable feedback. We provide our responses below:
>
> # Re. to R1
> Thanks for your valuable question, and we would like to provide the following explanation: Our work is primarily focused on scenarios that **emphasize reliability and accuracy**. We aim to strike a balance between reliability and computational cost by introducing a modest additional cost and directing the refinement overhead specifically to the most necessary parts—the unsure-labeled samples.
>
> To illustrate this, we conducted a detailed analysis of latency for the Llama3-8B-based results on the MultiWOZ-2.4 dataset. The table below presents the specific latency numbers (in seconds) for each method.
>
> | Method Type | Method | Inference_time | Refinement_rate | Refinement_time | Total_time | Final JGA |
> | --- | --- | --- | --- | --- | --- | --- |
> | Prompt | Direct | 354.63 | 0 | 0 | 354.63 | 6.73 |
> | Prompt | Direct | 354.63 | 100 | 4655.89 | 5010.52 | 36.01 |
> | Prompt | Self Reflection | 1500.93 | 55.43 | 2654.36 | 4155.29 | 29.72 |
> | Uncertainty | Self Consistency | 3529.82 | 69.26 | 3165.32 | 6695.14 | 20.7 |
> | Reliability(ours) | SFT + DPO Post | 559.03 | 27.69 | 1684.27 | 2243.3 | 56.33 |
>
> For the Direct Method, we reported the latency for scenarios with no refinement and full refinement. In contrast, some traditional reliability-enhancing methods—such as Self Reflection and Self Consistency—incur higher inference times due to additional reasoning or sampling steps. As can be seen from the table, our method achieves a notable improvement in reliability while introducing only a reasonable additional cost.

---

> ### Comment · Area_Chair_Ga6b · 2025-06-07
>
> Dear reviewer,
>
> thank you for your valuable effort. I noticed that you have not answered the authors rebuttal. Could please let us know what are your thoughts and if the answer has been satisfactory? Do you have additional questions?  Thank you!

---

### Official Review · Reviewer_MJZ5 · 2025-05-13

**Rating:** 7
**Confidence:** 3
**Ethics Flag:** 1

**Summary:**

Authors introduce a method for better dialogue state tracking (DST) in the context of task-oriented dialogue systems. The broader principle is to have models scrutinize their own outputs, rating their own certainty, and doing additional refinement on outputs that the model deems it's unsure of. Although tested only on DST in the context of this paper, the principle may have wider applicability. The method the authors propose is composed of two components:

a) additional training to increase accuracy of sure/unsure self-eval (via both SFT and DPO)
b) training a refinement model to refine the unsure responses

Authors hope to add additional nuance to "classical" selective prediction (where the model either predicts or abstains) by transforming unsure outputs into something that is still partially useful for the user.

**Questions To Authors:**

Fig. 2 "Resoning steps" -- typo

**Reasons To Accept:**

Method is validated quite comprehensively on DST datasets.

Empirical performance gains are significant and clearly demonstrated.

Broader principle is interesting.

**Reasons To Reject:**

Authors make implicit claims about broad applicability yet only test on DST. I think some of the claims/framing needs to be toned down.

Refinement model training is essentially a distillation of GPT-4o reasoning traces, of a closed model which limits reproducibility.

---

> ### Author Response · Authors · 2025-06-02
>
> # Re. to R2
> Thank you for this important comment. Our initial motivation for using GPT-4o was to maximize performance. However, as discussed in Section 4.6.3 of the paper, we compared our refinement model to other publicly available refinement models and found that the open-source DeepSeek-R1-70B model also demonstrated strong refinement capabilities compared to GPT-4o.
>
> To further address the reproducibility concern, we trained a new refinement model using the reasoning traces of DeepSeek-R1-Distill-Llama-70B. We applied this new refinement model to refine the predictions of the aligned Llama3-8B models on the MultiWOZ-2.4 dataset, and the results are as shown in the following table. Specifically, these results reflect the overall Joint Goal Achievement (JGA) after refining the same "unsure" predictions with both versions of the refinement model and merging them with the "sure" predictions, thereby providing insight into the performance of the refinement model.
>
> | Method \ Reasoning traces source | GPT4 | DeepSeek-R1-Distill-Llama-70B |
> | --- | --- | --- |
> | SFT | 53.76 | 54.25 |
> | SFT + DPO Joint | 54.31 | 55.65 |
> | SFT + DPO Post | 56.33 | 57.69 |
>
> Upon analyzing the performance of the DeepSeek-R1-Distill-Llama-70B-based refinement model, we found that it slightly outperformed the GPT-4-based version. Moreover, by examining the generated reasoning traces, we observed that the DeepSeek-based refinement process tended to produce longer and more detailed steps, which may have contributed to the improved accuracy.
>
> # Re. to Q1
> Thank you very much for catching this typo. We will correct it in the final version.

---

> > ### Comment · Reviewer_MJZ5 · 2025-06-07
> >
> > Thank you for confirming that this works with other models providing the Chain of Thought (not GPT-4o) but the fact remains that this is effectively just distilling the CoT from the larger models? Is there not a disconnect, where you are tuning on the certainty of the large model (its reasoning traces where it concludes it's unsure for example), but this doesn't particularly make sense since the "knowledge" of the large model is not necessarily the same as the small? i.e. the small model should be much more conservative in its predictions given its smaller parameter count (like it "knows fewer things") but this is not necessarily what it's learning from the larger model, whose traces are based on its own knowledge.

---

> > > ### Author Response · Authors · 2025-06-08
> > >
> > > We apologize if our presentation may have caused any confusion regarding the role of the refinement model. We further clarify the methodology proposed in our paper:
> > >
> > > Our approach consists of two main stages. The first stage focuses on aligning the model’s self-awareness capabilities. Specifically, we employ Supervised Fine-Tuning (SFT) and Direct Preference Optimization (DPO) to enable the model to explicitly label its predictions as “sure” or “unsure.” This stage is centered on calibrating the model’s **knowledge boundaries** and represents the core contribution of our paper.
> > >
> > > The second stage involves applying an external, high-performing refinement model to revise the predictions labeled as “unsure” by the aligned model from the first stage. This can be viewed as an auxiliary enhancement step. The refinement model can be chosen flexibly and is not required to maintain the same knowledge boundary as the aligned model in the first stage. In fact, we expect the refinement model to have higher accuracy, which is why we distill reasoning traces from a stronger model such as GPT-4o.
> > >
> > > A concrete example illustrating both stages is provided below.
> > >
> > > ```
> > > Input Dialog history:
> > > USER: Hi, I'm looking for a hotel to stay in.
> > > ....
> > > USER: Okay, please book that for 3 people and 2 nights starting from Friday.
> > > SYSTEM: Booking was successful. Reference number is : 9HMD04UW. anything else?
> > > USER: I would love to find a restaurant as well.
> > > ...
> > >
> > > Stage1 predictions:
> > > {
> > >     "hotel": {
> > >         "bookday": {
> > >             "value": "friday",
> > >             "confidence": "sure"
> > >         },
> > >         "bookpeople": {
> > >             "value": "3",
> > >             "confidence": "sure"
> > >         },
> > >         "bookstay": {
> > >             "value": "2",
> > >             "confidence": "sure"
> > >         }
> > >     },
> > >     "restaurant": {
> > >         "bookpeople": {
> > >             "value": "3",
> > >             "confidence": "unsure"
> > >         }
> > >     }
> > > }
> > >
> > > Stage2 refinement for "unsure" predictions:
> > > Target: {'domain': 'restaurant', 'slot': 'bookpeople', 'value': '3'}
> > >
> > > Refinement model:
> > > Thinking steps:
> > > step1: structure aspect: "bookpeople" is one of the slots in the restaurant domain and the value is specified as "3".
> > > step2: semantic aspect: In this case, the user never specifies the number of people for the restaurant booking. The number "3" is related to the hotel booking, not the restaurant. Therefore, the slot-value pair is irrelevant to the user's intent for the restaurant and should be removed.
> > > step3: the predicted slot-value pair is incorrect in semantic, and the slot should be removed.
> > > Refine action: removing
> > > Refined slot-value pair: none
> > > ```

---

> > > > ### Comment · Reviewer_MJZ5 · 2025-06-09
> > > >
> > > > Thank you for the clarifications on methodology, I did a deeper dive into the details in the appendix and I understand the method better now. I do believe the presentation of the method in the main paper could be somewhat improved, but I believe the paper has more solid footing than I originally thought. Raising my score. Thank you for engaging with me on this.

---

> > > > > ### Author Response · Authors · 2025-06-10
> > > > >
> > > > > Thank you! We are truly grateful to your suggestions and the raised score. Your engagement and constructive feedback are greatly appreciated, and we will carefully revise the presentation in the main paper in the final version.

---

> ### Author Response · Authors · 2025-06-02
>
> Thanks for your detailed review and valuable feedback. We provide our responses below:
>
> # Re. to R1
> Thank you for this valuable comment. However, we would like to emphasize that our method indeed possesses strong generalization capabilities and broad applicability. To support this, we have further supplemented results demonstrating our method’s performance on mathematical tasks.
>
> ### (1) Reason for incorporating only DST-related experiments in the main paper
> The motivation of our work is to improve model reliability while addressing the limitations of existing approaches that rely on rejecting difficult queries—an approach that can compromise the model’s helpfulness. This is especially true in multi-output tasks, where it may be inappropriate to reject an entire sample if only part of the output is unreliable. We thus aimed to find a multi-output task and DST naturally meet this criterion. Additionally, since multi-output tasks are often evaluated using F1-based metrics, we designed our proposed metrics to align with this structure. For the sake of maintaining metric consistency throughout the paper, we did not include experiments on other tasks in the main text.
>
> ### (2) Supplementary experiments on a math task:
> We conducted experiments on the widely used mathematics dataset **GSM8K**, applying our alignment method and comparing it with the baseline methods mentioned in our paper. We conducted the experiments on the Qwen2.5-7B-Instruct model and used 1000 samples as training set. Our approach also demonstrated promising improvements, effectively enhancing the model’s self-assessment capability while maintaining reliability at a reasonable additional cost.
>
> For the GSM8K task, we evaluated the model under two different test settings: (a) using Chain-of-Thought (CoT) reasoning and (b) directly predicting the final answer. LLMs generally perform differently under these two settings, so we tested both. For the refinement model, we used the DeepSeek-R1-Distill-Llama-70B model, which has strong mathematical reasoning capabilities.
>
> The results are summarized in the accompanying table, where the metric is adapted from the paper’s F1-based definitions to accuracy-based definitions:
>
> $$
> Weighted\ Accuracy(X,\alpha)\ = \frac{\sum\limits_{x_i\in Sure}S(x_i) + \alpha \times \sum\limits_{x_j\in Unsure}S(x_j)}{N_{Sure}+\alpha\times N_{Unsure}}
> $$
>
> $$
> Quality\ Accuracy(X)\ = Weighted\ Accuracy(X, 0.5)
> $$
>
> Here, the value of $S(x)$ is either 0 or 1, representing whether x is correct. The value of $\alpha$ is in the interval [0, 1].
>
> - **Quality_accuracy** assigns a weight of 0.5 to unsure samples.
> - **Final_accuracy** represents the model’s overall accuracy after refinement of "unsure" predictions.
>
> Results for experiment: Answer with CoT Reasoning:
>
> | Method Type | Method | Quality accuracy | Final accuracy |
> | --- | --- | --- | --- |
> | Prompt | Direct | 86.81 | 86.81 |
> | Prompt | Verbose | 90.8 | 90.98 |
> | Uncertainty | Prob. | 87.69 | 88.55 |
> | Uncertainty | Self Consistency | 89.41 | 91.48 |
> | Reliability(ours) | SFT | 89.69 | 91.05 |
> | Reliability(ours) | SFT + DPO Joint | 90.14 | 93.1 |
> | Reliability(ours) | SFT + DPO Post | 92.73 | 95.3 |
>
> Results for experiment: Answer without CoT Reasoning:
> | Method Type | Method | Quality accuracy | Final accuracy |
> | --- | --- | --- | --- |
> | Prompt | Direct | 19.94 | 19.94 |
> | Prompt | Verbose | 21.44 | 24.34 |
> | Uncertainty | Prob. | 24.49 | 76.5 |
> | Uncertainty | Self Consistency | 24.37 | 70.05 |
> | Reliability(ours) | SFT | 27.31 | 77.94 |
> | Reliability(ours) | SFT + DPO Joint | 26.16 | 78.24 |
> | Reliability(ours) | SFT + DPO Post | 27.83 | 79.83 |
>
> In both evaluation settings, our alignment method significantly outperformed both prompt-based and uncertainty-based baselines in improving the model’s self-assessment ability—specifically, the ability to accurately label predictions as sure or unsure during inference. Our method achieved higher Quality-Accuracy compared to the baselines, with the accuracy of predictions labeled as certain being markedly higher than that of the unsure-labeled answers. It is important to note that **the model's inherent mathematical performance did not improve**: the accuracy of predictions without labels was nearly identical to that of the original Qwen2.5-7B-Instruct model. In other words, the enhancement of Quality-F1 is entirely attributable to the improved self-awareness capability of the model. Moreover, after applying refinement with the same refine model, our method achieved higher final accuracy at a reasonable cost.

---

> > ### Comment · Reviewer_MJZ5 · 2025-06-07
> >
> > I'm not quite sure I understand how this method is applied to GSM8K. There is no "multi-output" component here that you can modulate based on certainty, no? So this is essentially just selective prediction?

---

> > > ### Author Response · Authors · 2025-06-08
> > >
> > > Thank you for your time and thoughtful questions. Our further clarifications are as follows:
> > >
> > > (1) Regarding GSM8K, it is true that we did not include a “multi-output” component in this task. This is because we consider Dialog State Tracking (DST) to be a well-established multi-output task. In our paper, we have already conducted extensive experiments on multiple DST benchmarks to demonstrate the effectiveness of our method in multi-output settings. To further validate the generality of our approach, we extended our method to the classic single-output mathematical reasoning benchmark GSM8K during the rebuttal phase.
> > >
> > > (2) On GSM8K, our method is not simply performing selective prediction. Instead, it still attempts to answer all questions while also estimating confidence for each prediction. We adapt the F1 metric used for evaluating multi-output tasks to the accuracy metric suitable for single-output tasks. A specific example of the model’s output is shown below.
> > >
> > > ```
> > > Question:
> > > In 21 hours, a heating plant consumes 7 liters of fuel. How much does it consume in 90 hours?
> > >
> > > Answer:
> > > Let's think step by step: To find out how much fuel the heating plant consumes in 90 hours, we need to follow these steps:
> > > Step 1: Calculate the hourly fuel consumption rate.
> > > - The heating plant consumes 7 liters of fuel in 21 hours.
> > > - To find out how much fuel it consumes per hour, we divide the total fuel consumed by the total hours: \\( \\frac{7 \\text{ liters}}{21 \\text{ hours}} = \\frac{1}{3} \\text{ liters per hour} \\).
> > > Step 2: Multiply the hourly consumption rate by 90 hours.
> > > - Now that we know the heating plant consumes \\( \\frac{1}{3} \\) liters per hour, we can find out how much it consumes in 90 hours by multiplying: \\( \\frac{1}{3} \\text{ liters per hour} \\times 90 \\text{ hours} = 30 \\text{ liters} \\).
> > > The final answer is: 90.
> > > Confidence: unsure.
> > > ```

---

### Official Review · Reviewer_GHeu · 2025-05-13

**Rating:** 6
**Confidence:** 3
**Ethics Flag:** 1

**Summary:**

The work tries to make LLM generations more reliable and less hallucinated by generating "sure" and "unsure" labels. The unsure labels then undergo a COT refinement process. The method first labels correct responses as "sure" and incorrect as "unsure". Then performs DPO on after collecting the training data. It then uses a refinement model to correct the output for "unsure" labelled generations.

**Questions To Authors:**

- Are all the performance gains statistically significant?
- Any intuition why the improvement with SGD is low?
Can you look at the logits for sure and unsure to better understand? There can be cases where the model spits sure, but the probability is just marginally higher than unsure.
- \`\` '' for quotes in latex.
- Can you apply this method for a more meaningful task?

**Reasons To Accept:**

- Novel technique for improving alignment in language models.
- Performs extensive evaluation and shows that the method has its merit.

**Reasons To Reject:**

- The motivation for this problem is unclear. The motivation and the examples presented in the paper can be solved by providing specs for the model. In Figure 1, why is 7:59 not a valid answer?
- Although the work talks about using sampling for collecting sure and unsure labels, the work samples only one generation and labels all the correct output as "sure" and all the incorrect ones as "unsure". This goes against their motivation of having "unsure" for correct and incorrect generations.

---

> ### Author Response · Authors · 2025-06-02
>
> # Re. to Q4
> Thank you for your insightful question. Below, we provide our explanation and supplementary experimental results on a more meaningful task—mathematics reasoning.
>
> ### (1) Why we used the DST task
> The motivation behind our paper is to enhance model reliability while mitigating the limitations of existing rejection-based approaches, which often involve rejecting difficult problems outright. Such approaches can compromise the model’s helpfulness, especially in multi-output tasks, where rejecting an entire sample can be inappropriate. Therefore, we aimed to identify a multi-output task where the model has high confidence in part of its outputs while being less confident in others. The DST task naturally meets this criterion. Additionally, as multi-output tasks typically use F1-based evaluation metrics, and our proposed metrics are also based on F1, we did not include more tasks in the main paper to maintain consistency in evaluation metrics.
>
> ### (2) Supplementary experiments on a math task
> We conducted experiments on the widely used mathematics dataset **GSM8K**, applying our alignment method and comparing it with the baseline methods mentioned in our paper. We conducted the experiments on the Qwen2.5-7B-Instruct model and used 1000 samples as training set. Our approach also demonstrated promising improvements, effectively enhancing the model’s self-assessment capability while maintaining reliability at a reasonable additional cost.
>
> For the GSM8K task, we evaluated the model under two different test settings: (a) using Chain-of-Thought (CoT) reasoning and (b) directly predicting the final answer. LLMs generally perform differently under these two settings, so we tested both. For the refinement model, we used the DeepSeek-R1-Distill-Llama-70B model, which has strong mathematical reasoning capabilities.
>
> The results are summarized in the accompanying table, where the metric is adapted from the paper’s F1-based definitions to accuracy-based definitions:
> $$
> Weighted\ Accuracy(X,\alpha)\ = \frac{\sum\limits_{x_i\in Sure}S(x_i) + \alpha \times \sum\limits_{x_j\in Unsure}S(x_j)}{N_{Sure}+\alpha\times N_{Unsure}}
> $$
>
> $$
> Quality\ Accuracy(X)\ = Weighted\ Accuracy(X, 0.5)
> $$
>
> Here, the value of $S(x)$ is either 0 or 1, representing whether x is correct. The value of $\alpha$ is in the interval [0, 1].
>
> - **Quality_accuracy** assigns a weight of 0.5 to unsure samples.
> - **Final_accuracy** represents the model’s overall accuracy after refinement of "unsure" predictions.
>
> Results for experiment: Answer with CoT Reasoning:
>
> | Method Type | Method | Quality accuracy | Final accuracy |
> | --- | --- | --- | --- |
> | Prompt | Direct | 86.81 | 86.81 |
> | Prompt | Verbose | 90.8 | 90.98 |
> | Uncertainty | Prob. | 87.69 | 88.55 |
> | Uncertainty | Self Consistency | 89.41 | 91.48 |
> | Reliability(ours) | SFT | 89.69 | 91.05 |
> | Reliability(ours) | SFT + DPO Joint | 90.14 | 93.1 |
> | Reliability(ours) | SFT + DPO Post | 92.73 | 95.3 |
>
> Results for experiment: Answer without CoT Reasoning:
> | Method Type | Method | Quality accuracy | Final accuracy |
> | --- | --- | --- | --- |
> | Prompt | Direct | 19.94 | 19.94 |
> | Prompt | Verbose | 21.44 | 24.34 |
> | Uncertainty | Prob. | 24.49 | 76.5 |
> | Uncertainty | Self Consistency | 24.37 | 70.05 |
> | Reliability(ours) | SFT | 27.31 | 77.94 |
> | Reliability(ours) | SFT + DPO Joint | 26.16 | 78.24 |
> | Reliability(ours) | SFT + DPO Post | 27.83 | 79.83 |
>
> In both evaluation settings, our alignment method significantly outperformed both prompt-based and uncertainty-based baselines in improving the model’s self-assessment ability—specifically, the ability to accurately label predictions as sure or unsure during inference. Our method achieved higher Quality-Accuracy compared to the baselines, with the accuracy of predictions labeled as certain being markedly higher than that of the unsure-labeled answers. It is important to note that **the model's inherent mathematical performance did not improve**: the accuracy of predictions without labels was nearly identical to that of the original Qwen2.5-7B-Instruct model. In other words, the enhancement of Quality-F1 is entirely attributable to the improved self-awareness capability of the model. Moreover, after applying refinement with the same refine model, our method achieved higher final accuracy at a reasonable cost.

---

> > ### Comment · Reviewer_GHeu · 2025-06-03
> >
> > I am satisfied with the author's responses. I would raise my score to 6, but I strongly recommend that the authors resubmit the work with all the improvements. It will make for a better paper.

---

> > > ### Author Response · Authors · 2025-06-04
> > >
> > > Thank you! We are very grateful to your encouraging comments and the increased score. Your detailed questions and suggestions are very helpful and we will carefully incorporate the recommended improvements into the final version of the paper.

---

> ### Author Response · Authors · 2025-06-02
>
> # Re. to Q2
> ### (1) Intuition for the Low Improvement with SGD
>
> Thank you for your insightful observation. As we understand it, your comment about the "**improvement with SGD is low**" refers to the self-awareness performance (i.e., the ability to correctly distinguish sure and unsure predictions) in Table 1 of the paper. We have conducted a more detailed analysis of this result.
>
> Specifically, we believe that the model is indeed capable of correctly identifying sure and unsure samples. To illustrate this, we selected one representative baseline method from the three types of baselines based on Llama3-8B, and compared it with our method in terms of Quality-F1 (which assigns a weight of 0.5 to unsure samples and can measure self-awareness), as well as the precision of sure and unsure predictions, and the overall proportion of sure predictions. The results are summarized in the table below. As shown, our method achieves significantly better Quality-F1 scores. Additionally, the precision of the sure-labeled samples is significantly higher than that of the unsure-labeled samples, further validating the model’s capability to distinguish sure from unsure predictions. The large difference in sure/unsure precision for Self Consistency is primarily due to its tendency to over-label predictions as unsure.
>
> | Method Type | Method | Quality F1 | Precision_sure | Precision_unsure | Sure_rate |
> | --- | --- | --- | --- | --- | --- |
> | Prompt | Self Reflection | 21.30 | 65.32 | 37.55 | 63.80 |
> | Uncertainty | Self Consistency | 16.55 | 84.66 | 20.78 | 46.84 |
> | Reliability(ours) | SFT + DPO Post | 64.14 | 89.07 | 30.58 | 81.35 |
>
> ### (2) Analysis of the Lower JGA after Refinement on SGD
>
> We also noticed that, on the SGD dataset, the overall Joint Goal Accuracy (JGA) after refinement is lower compared to the other two datasets. We conducted an analysis to understand the underlying causes, and we believe there are two primary reasons for this:
>
> 1. **Lower refinement model accuracy.** The SGD dataset itself is more challenging, with inconsistencies between the predefined slot spaces of the training and test sets, leading to higher refinement difficulty. We found that our refinement model’s average accuracy on SGD was only 74.63%, considerably lower than that on the other two datasets (MultiWOZ-2.4: 86.23%, BiTOD: 87.64%).
> 2. **Self-awareness performance of the reliability-aligned model.** Although the stage-one aligned model also demonstrates strong self-awareness on the SGD dataset, its performance is relatively lower compared to the MultiWOZ-2.4 and BiTOD datasets. For example, in the Llama3-8B based setting, the best sure-prediction precision was 89.07% on SGD, compared to 94.52% on MultiWOZ and 95.84% on BiTOD. Errors in this stage-one self-awareness prediction are retained and ultimately affect the final JGA performance after refinement.
>
> # Re. to Q3
> Thank you very much for pointing out this typographical issue. We will correct it in the final version.

---

> ### Author Response · Authors · 2025-06-02
>
> Thanks for your detailed review and valuable feedback. We provide our responses below:
>
> # Re. to R1
> We apologize for any misunderstanding that resulted from our previous provided example, which may have caused confusion.
> ### (1) Further clarification of the motivation behind the Dialog State Tracking (DST) task.
>
> The primary objective of DST is to track the dialogue state between a user and the system, which involves maintaining a set of predefined slots and assigning appropriate values to them when the user expresses relevant intents. Therefore, DST focuses on capturing the user’s direct intention.
>
> For example, in Figure 1, the time "7:00" is explicitly mentioned by the user and reflects the user’s actual preference. In contrast, "7:59" is an alternative suggested by the system. Since this suggestion may or may not be accepted by the user, DST should retain "7:00" as the accurate slot value.
>
> ### (2) A more specific and representative example.
> We have selected another example to better illustrate the requirements of the DST task—as well as the potential errors that may occur in existing systems—as shown below.
> ```
> Dialog history:
> USER: I am looking for a restaurant and book a table for 4 on Saturday at 12:45.
> SYSTEM: ......
> USER: I also need a train please.
> SYSTEM: ......
>
> Model Extracted Dialog State:
> {
> 	"restaurant": {
> 			...
> 			"bookpeople": "4",
> 			"day": "Saturday",
> 			"booktime": "12:45"
> 	},
> 	"train": {
> 			...
> 			"day": "Saturday"
> 	}
> }
> ```
> An incorrect extracted slot-value pair: **train: day = Saturday**
>
> Reason: Although the user explicitly mentioned he'd like to book a restaurant for Saturday, he didn't mention that he'd like to book a train ticket for the same Saturday. This is a slot-value pair obtained through erroneous inference and needs to be removed.
>
> # Re. to R2
> Thank you very much for raising this important question. In fact, we have addressed the number of sampling times in lines 165–167 of the paper. In our proposed method, any slot that exhibits inconsistent predictions across multiple samplings (such as the destination of a train) is labeled as *unsure*.
>
> ### (1) Our rationale for constructing the Supervised Fine-Tuning (SFT) dataset in this way
>
> If the model consistently produces correct predictions across all samplings, it indicates that this aspect is within the model’s capability range and should thus be labeled as *sure*. Conversely, if errors occur, it suggests that the prediction may be outside the model’s reliable capability, warranting an *unsure* label. We hope that such an SFT dataset can implicitly teach the model to learn self-assessment capabilities.
>
> ### (2) The relationship between sampling times and model behavior
>
> As the number of sampling times increases, the likelihood of encountering prediction errors also grows, resulting in a higher proportion of slots labeled as *unsure*—this leads to a more conservative model. Therefore, we currently choose a sampling count of one, which most accurately reflects the model’s inherent prediction ability without making it overly conservative. However, in scenarios where higher reliability is desired, the sampling count can be increased to prompt the model to classify more uncertain predictions as *unsure*.
>
> # Re. to Q1
> Thank you for raising this important question. Based on your suggestion, we conducted statistical significance tests on the results of Qwen2.5-7B on the DST dataset, specifically evaluating the four metrics presented in Table 1 (Precision_sure, Recall_total, Optimal-F1, and Quality-F1). We performed both **paired *t*-tests** and **Wilcoxon signed-rank tests** to compare our method (SFT+DPO Joint) with six baseline methods.
>
> The obtained *p*-values for all tests were well below the conventional threshold of 0.01. This strongly indicates that the performance gains of our method are statistically significant compared to the baseline methods.

---

### Decision · Program_Chairs · 2025-07-08

**Decision:**

Accept

**Comment:**

This paper introduces the Explicit Knowledge Boundary Modeling (EKBM) framework, a two-stage process designed to improve the reliability of LLMs and reduce hallucinations. The system first uses a "fast-thinking" model to generate answers and label each part of the output with a confidence level, either "sure" or "unsure." High-confidence "sure" outputs can be used immediately, while "unsure" outputs are passed to a second, "slow-thinking" refinement model for more deliberate analysis and correction. The framework was tested on dialogue state tracking (DST) tasks, showing that it improves the model's ability to assess its own accuracy and boosts overall performance.

Overall, the agreement among reviewers is that the paper has a sound experimental setup, makes substantial contributions to the scientific community, and presents solid empirical results.Upon the reviewers' request, the authors further show that the performance improvements are statistically significant. Weaknesses identified by the reviewers included a narrow experimental setup focused on DST tasks, but the authors successfully addressed this by expanding the experiments to mathematical reasoning. The authors also extended their findings to open-source models (from the initial GPT-centric experiments). Other issues, such as clarity and paper presentation, can be easily addressed for the camera-ready version.

Overall, I agree with the reviewers' recommendation to accept this paper.

[Automatically added comment] At least one review was discounted during the decision process due to quality]